# A shared genetic contribution to breast cancer and schizophrenia

Donghao Lu 1,2,3,11✉, Jie Song 1,11✉, Yi Lu1, Katja Fall4, Xu Chen 1, Fang Fang5, Mikael Landén1,6, Christina M. Hultman1,7, Kamila Czene1, Patrick Sullivan1,8, Rulla M. Tamimi2,3,9,12 & Unnur A. Valdimarsdóttir 1,3,10,12

An association between schizophrenia and subsequent breast cancer has been suggested; however the risk of schizophrenia following a breast cancer is unknown. Moreover, the driving forces of the link are largely unclear. Here, we report the phenotypic and genetic positive associations of schizophrenia with breast cancer and vice versa, based on a Swedish population-based cohort and GWAS data from international consortia. We observe a genetic correlation of 0.14 (95% CI 0.09–0.19) and identify a shared locus at 19p13 (*GATAD2A*) associated with risks of breast cancer and schizophrenia. The epidemiological bidirectional association between breast cancer and schizophrenia may partly be explained by the genetic overlap between the two phenotypes and, hence, shared biological mechanisms.

[1] Department of Medical Epidemiology and Biostatistics, Karolinska Institutet, Nobels väg 12A, 17177 Stockholm, Sweden. [2] Channing Division of Network Medicine, Brigham and Women's Hospital, Harvard Medical School, 181 Longwood Avenue, Boston, MA 02115, USA. [3] Department of Epidemiology, Harvard T.H. Chan School of Public Health, 677 Huntington Avenue, Boston 02115 MA, USA. [4] Clinical Epidemiology and Biostatistics, School of Medical Sciences, Örebro University, Campus USÖ, 70182 Örebro, Sweden. [5] Unit of Integrative Epidemiology, Institute of Environmental Medicine, Karolinska Institutet, Nobels väg 13, 17177 Stockholm, Sweden. [6] Department of Psychiatry and Neurochemistry, Institute of Neuroscience and Physiology, The Sahlgrenska Academy, University of Gothenburg, Sahlgrenska University Hospital, Blå stråket 15, 41345 Gothenburg, Sweden. [7] Department of Psychiatry, Icahn School of Medicine, Mt. Sinai Hospital, 1 Gustave L. Levy Place, New York, NY 10029, USA. [8] Departments of Genetics and Psychiatry, University of North Carolina, 120 Mason Farm Road, Chapel Hill, NC 27599, USA. [9] Department of Healthcare Research and Policy, Weill Cornell Medicine, 402 East 67th Street, New York, NY 10065, USA. [10] Center of Public Health Sciences, Faculty of Medicine, University of Iceland, Sturlugata 8, 101 Reykjavik, Iceland. [11] These authors contributed equally: Donghao Lu, Jie Song. [12] These authors jointly supervised this work: Rulla M. Tamimi, Unnur A. Valdimarsdóttir. ✉email: donghao.lu@ki.se; jie.song@ki.se

reast cancer is the most commonly diagnosed cancer among women worldwide[1]. Some evidence suggests that vulnerability to stress[2–4], particularly indicated by the presence of psychiatric disorders including schizophrenia[5,6], may be associated with increased risk of subsequent breast cancer. Conversely, recent studies by us and others support that patients with breast cancer are at higher risk of a range of psychiatric disorders subsequently, including depression, anxiety, and post-traumatic stress disorder[7,8]. The longitudinal bidirectional association raises the question of shared risk factors between breast cancer and psychiatric disorders.

A hypothesized link between schizophrenia and breast cancer dates several decades back[9]. Affecting up to 1% of the population, schizophrenia is one of the most severe and costliest psychiatric disorders[10]. Several risk factors for breast cancer are commonly present in schizophrenia, including behavioral risk factors such as smoking, alcohol use, excess bodyweight, and physical inactivity[11–16], as well as prenatal- and early-life exposure to stress[17,18]. Despite lower attendance to mammography screening in patients with schizophrenia[19], the results from a recent meta-analysis suggest that schizophrenia is associated with increased risk of breast cancer[6]. Meanwhile, the subsequent risk of schizophrenia among breast cancer patients remains unclear. If confirmed, such longitudinal bidirectional association would lend further support to the shared risk factors between both diseases rather than one causing the other. Although recent studies suggest that genetic markers of schizophrenia are associated with higher risk of breast cancer[20,21], a potential genetic overlap between the diseases has not been thoroughly assessed.

Here we illustrate the longitudinal association of schizophrenia with subsequent risk of breast cancer and vice versa, by utilizing a Swedish population-based cohort. Leveraging data from international consortia, we further find evidence for a shared genetic basis between breast cancer and schizophrenia and identify a shared locus. Together with the pathways enriched in the functional annotation, our findings may shed light on the potential biological mechanisms that drive the susceptibility to both diseases.

## Results

**Longitudinal phenotypic association between two diseases.** Using nationwide population and health registers in Sweden, we identified all women with invasive breast cancer ($N = 94,626$) diagnosed in Sweden during 1990–2009 (mean age at diagnosis, $63.6 \pm 14.1$ years). For each cancer case, at the time of cancer diagnosis, we randomly selected 30 cancer-free women as controls ($N = 2,838,765$) who were individually matched on birth year. By cross-linkage to the Patient Register, we identified any first-ever inpatient or outpatient diagnoses of schizophrenia before or after the date of cancer diagnosis for cases and matching date for controls.

First, we examined the association of schizophrenia with subsequent invasive breast cancer. Five hundred and thirty-four (0.56%) and 11,036 (0.39%) inpatient diagnoses of schizophrenia were observed among women who later did vs. did not develop breast cancer, respectively. The nested case–control analysis (i.e., assessing the incidence of schizophrenia before the cancer diagnosis/matching date), by design, is equivalent to the prospective analysis using full cohort[22]. Schizophrenia was associated with a 49% increased risk of subsequent invasive breast cancer (95% confidence interval [CI], 37–63%, $P = 1.72 \times 10^{-19}$; Table 1). The association remained robust when including outpatient diagnoses of schizophrenia and/or additionally controlling for characteristics shared between breast cancer and schizophrenia, such as parity, and previous diagnoses of

psychiatric disorders (including substance use disorders) and obesity (Table 1).

Second, we assessed the risk of schizophrenia following invasive breast cancer in a matched cohort analysis (i.e., addressing incidence of schizophrenia after the cancer diagnosis/matching date). We excluded women with inpatient or outpatient diagnosis of schizophrenia prior to the matching ($N = 11,947$). During the follow-up (from the matching up to December 31, 2010), we noted 100 (0.11%) and 2584 (0.09%) first-ever inpatient diagnoses of schizophrenia among women with and without breast cancer, respectively. We found that women with invasive breast cancer were at 31% increased risk of hospitalization for schizophrenia (95% CI, 7–61%, $P = 0.008$). An increased risk was also indicated when we included outpatient visits for schizophrenia and/or additionally controlling for parity and previous diagnoses of psychiatric disorders and obesity (Table 1).

To shed light on the early-onset schizophrenia, we restricted the analysis to women aged 18–44 years and suggested a bidirectional, positive association between early-onset schizophrenia and breast cancer, although the numbers were small and estimates were not statistically significant (Supplementary Table 1).

These results confirm a longitudinal, bidirectional association between breast cancer and schizophrenia, which is not entirely explained by proxies of major environmental risk factors shared by both traits. Although unmeasured confounders cannot be ruled out, one possible explanation is the common genetic factors underlying both traits.

**Genetic overlap between breast cancer and schizophrenia.** Given the observation in the epidemiological analysis, we subsequently focused on confirming and identifying genetic overlap between schizophrenia and breast cancer in the genetic analysis. First, we calculated the genetic correlation between breast cancer and schizophrenia using linkage disequilibrium (LD) score regression. Genetic correlation is the proportion of variance that two traits share due to common genetic factors, as an estimate of the degree of pleiotropy or genetic overlap.

Using summary statistics for breast cancer[23] (122,977 cases and 105,974 controls) and schizophrenia[24] (33,640 cases and 43,456 controls) from the consortia's meta-analyzed genome-wide association studies (GWASs) in populations of European ancestry, we estimated the genetic correlation between both traits at 0.14 (95% CI, 0.09–0.19, $P = 5.30 \times 10^{-8}$), suggesting that around one-seventh of the genetic contribution to these two phenotypes is shared. A significant, albeit small, LD score regression intercept (0.03, 95% CI 0.01–0.04) suggests a minor sample overlap between both consortia, which is largely inevitable.

Based on the same GWAS summary statistics, we next calculated polygenic risk scores (PRSs) to further assess the genetic association between breast cancer and schizophrenia. Single-nucleotide polymorphisms (SNPs) in the extended major histocompatibility complex (MHC) region (chr6: 25–34 Mb) were removed due to the long-range LD and unusual genetic architecture. The PRS for schizophrenia was significantly associated with the risk of breast cancer and vice versa across most $P$ value thresholds ($P_T$s; Fig. 1). The association became more significant along with increasing threshold. At $P_T < 0.4712$ (i.e., PRS was computed using all SNPs below the threshold after LD clumping), the association between PRS for schizophrenia and breast cancer yielded the lowest $P$ value (odds ratio [OR] 1.023, 95% CI 1.022–1.025, $P = 3.57 \times 10^{-186}$), explaining 0.37% of the variance. At $P_T < 0.4492$, the association between PRS for

**Table 1 Longitudinally bidirectional association between invasive breast cancer and schizophrenia.**

| | Women without breast cancer | Women with breast cancer | | | | |
|---|---|---|---|---|---|---|
| | N (%) | N (%) | OR (95% CI)[a] | P[a] | OR (95% CI)[b] | P[a] |
| **Association of schizophrenia with subsequent breast cancer[c]** | | | | | | |
| Number of women | 2,838,765 | 94,626 | — | — | — | — |
| Clinical diagnosis of schizophrenia | | | | | | |
| Inpatient diagnosis | 11,036 (0.39) | 534 (0.56) | 1.49 (1.37–1.63) | $1.72 \times 10^{-19}$ | 1.40 (1.28–1.53) | $4.28 \times 10^{-14}$ |
| Inpatient or outpatient diagnosis | 11,399 (0.40) | 548 (0.58) | 1.48 (1.36–1.62) | $2.54 \times 10^{-19}$ | 1.39 (1.28–1.52) | $1.17 \times 10^{-14}$ |
| | N (%) | N (%) | HR (95% CI)[a] | P[a] | HR (95% CI)[b] | P[a] |
| **Association of breast cancer with subsequent schizophrenia[d]** | | | | | | |
| Number of women | 2,827,366 | 94,078 | — | — | — | — |
| Clinical diagnosis of schizophrenia | | | | | | |
| Inpatient diagnosis | 2584 (0.09) | 100 (0.11) | 1.31 (1.07–1.61) | 0.008 | 1.29 (1.03–1.62) | 0.029 |
| Inpatient or outpatient diagnosis | 3728 (0.13) | 135 (0.14) | 1.23 (1.03–1.46) | 0.019 | 1.13 (0.93–1.37) | 0.218 |

N number, OR odds ratio, HR hazard ratio.
[a]Models were adjusted for educational level (primary school, high school, college and beyond, or unknown) and region of residence (southern, central, or northern Sweden). Birth year and age at reference were inherently controlled for due to matching.
[b]Models were additionally adjusted for parity (0, 1–2, or ≥3), pre-existing psychiatric disorder (yes or no; including substance use disorders), and obesity (yes or no) at matching.
[c]Based on the nested case–control study. The estimates, i.e., OR derived from conditional logistic regression, should be interchangeably interpreted as the risk of breast cancer among patients with schizophrenia.
[d]Based on the matched cohort study. The estimates, i.e., HR derived from stratified Cox proportional hazards regression, are interpreted as the risk of schizophrenia among patients with breast cancer.

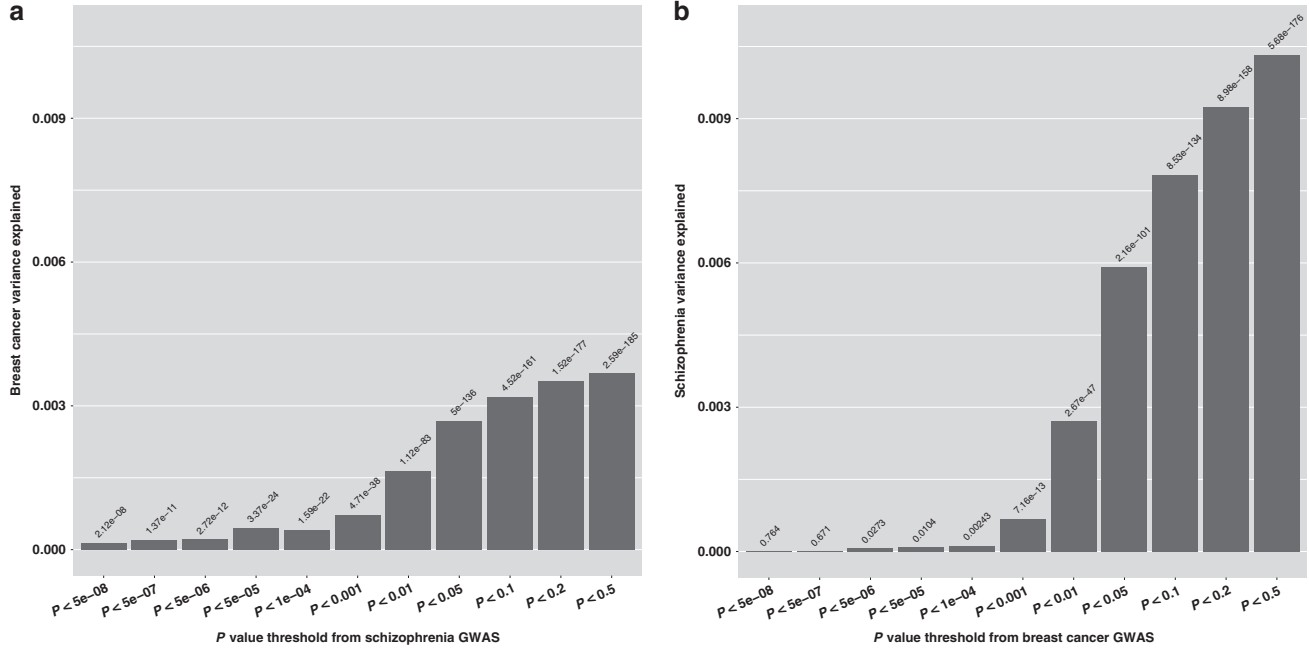

**Fig. 1 Genetic association between breast cancer and schizophrenia based on GWAS summary statistics of breast cancer and schizophrenia. a** PRS for schizophrenia associated with risk of breast cancer. **b** PRS for breast cancer associated with risk of schizophrenia. We performed PRS analysis based on GWAS summary statistics. We plotted the variance of one disease (Y axis) explained by the genetic markers associated with the other disease under a P value threshold (X axis). The number above the bar indicates the statistical significance of the genetic association (two-sided P value). GWAS genome-wide association study, PRS polygenic risk score.

breast cancer and schizophrenia yielded the lowest P value (OR 1.070, 95% CI 1.065–1.075, $P = 3.16 \times 10^{-177}$), explaining 1.04% of the variance. Notably, the association between the breast cancer PRS and schizophrenia appears when large numbers of SNPs weakly associated with breast cancer are included in the analysis, in support of the polygenic architecture in schizophrenia.

Circulating sex hormones are well-confirmed predictors of breast cancer risk[25], whereas sex hormones may possibly play a role in schizophrenia development, particularly the differential risk between women and men[26]. To address potential differences in the genetic overlap between different subgroups of breast cancer and schizophrenia, we performed separate analyses among estrogen receptor (ER)-positive and ER-negative breast cancer as well as female and male schizophrenia. Using the GWAS summary statistics from consortia, similar findings were noted in the associations between schizophrenia and both ER-positive and ER-negative breast cancer (Supplementary Fig. 1). As the

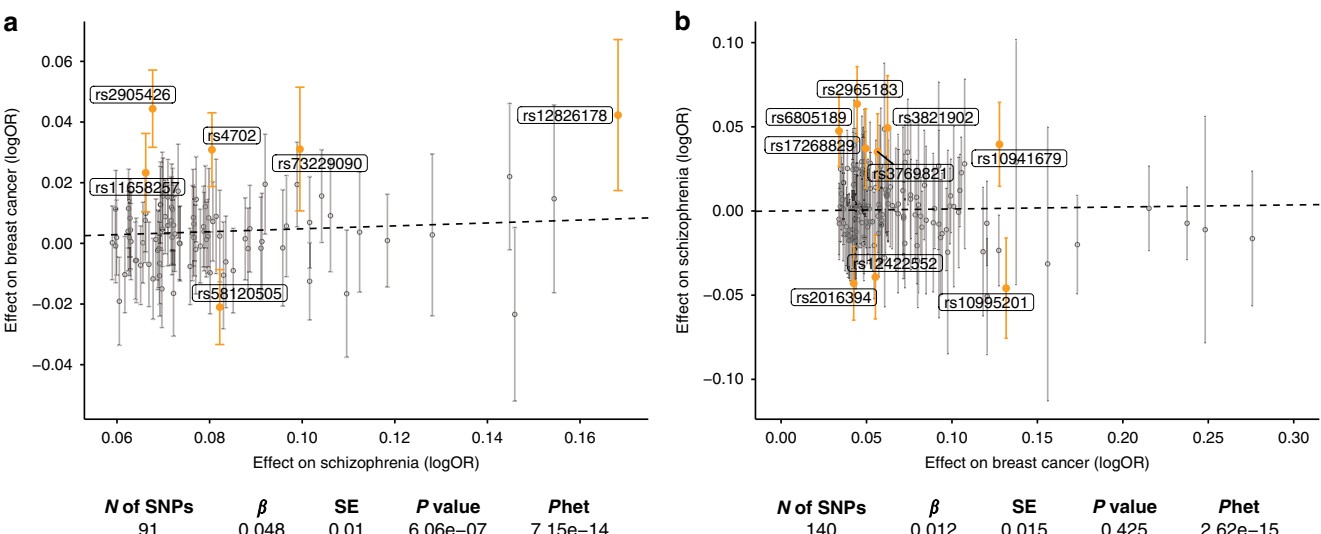

**Fig. 2 Associations of lead SNPs for schizophrenia with the risk of breast cancer, and vice versa. a** Lead SNPs for schizophrenia associated with breast cancer (n = 122,977 breast cancer cases and 105,974 controls). **b** Lead SNPs for breast cancer associated with schizophrenia (n = 36,989 schizophrenia cases and 113,075 controls). We plotted the associations of genetic markers for one disease (the effect size shown as X axis) with the other disease (Y axis) based on GWAS summary statistics. Data are presented as OR (dots) and 95% confidence interval (error bars). Orange dots denote variants of P < 0.05 after false discovery rate (FDR) adjustment. We then derived the average effect of the set of genetic markers on the other disease using a global test. The dash line indicates this effect size, while the full statistical results were reported below the plot. P values are two sided. N number, OR odds ratio; Phet, P for heterogeneity, SD standard deviation, SE standard error, SNP single-nucleotide polymorphism.

GWAS summary statistics are not publicly available for schizophrenia by sex, we then used individual-level genotype data from the Swedish Schizophrenia Study [S3] for such analysis (in total 4924 cases of schizophrenia and 6207 controls). Despite less significant associations compared with the results found using consortia data, similar magnitudes in effect sizes (i.e., ORs) of PRS for breast cancer were noted between female and male schizophrenia (Supplementary Table 2).

**A distinct signal shared by breast cancer and schizophrenia.** To identify the strongest genetic loci shared between breast cancer and schizophrenia, we first visualized the associations of lead SNPs (i.e., genome-wide significant SNPs after LD clumping) for schizophrenia with the risk of breast cancer, and vice versa. When assessing 91 genome-wide significant SNPs for schizophrenia, we approximated the average effects of these alleles on breast cancer and observed a positive association ($P = 6.07 \times 10^{-6}$; Fig. 2). When focusing on the 140 lead SNPs for breast cancer, we observed no overall correlation with schizophrenia ($P = 0.425$). After correcting for multiple testing, six lead SNPs of schizophrenia were significantly associated with breast cancer, while nine lead SNPs of breast cancer were also associated with schizophrenia.

Considering the significant heterogeneity in the tests of association, we further visualized the associations by filtering all influential points out (according to Cook's distance) and by removing SNPs iteratively until the heterogeneity test was no longer significant. The results remained largely similar. The association between lead SNPs for schizophrenia and risk of breast cancer was significant ($P = 0.024$), although somewhat attenuated (Supplementary Fig. 2).

The 15 lead SNPs come from 14 loci, which we examined individually by using regional association plots to assess colocalization of the signals (i.e., if the top variants for the 2 traits are highly correlated). We identified a common locus at 19p13 anchored by the lead SNPs rs2965183 and rs2905426 (LD $r^2 = 0.93$) for breast cancer and schizophrenia, respectively (Fig. 3

for locus 19p13 and Supplementary Fig. 3 for other loci). The variants at 19p13 was positively associated with increased risks of both breast cancer and schizophrenia (e.g., rs2965183: effect size = 0.04, $P = 8.40 \times 10^{-12}$ for breast cancer; effect size = 0.07, $P = 4.07 \times 10^{-9}$ for schizophrenia). Based on PhenoScanner database[27], this locus is also associated with lipid levels (including total cholesterol (TC), triglycerides (TG), and low-density lipoprotein [LDL]). However, the associations with lipid levels are in the opposite direction as seen for breast cancer and schizophrenia (Supplementary Table 3). Moreover, we plotted regional associations of locus 19p13 with lipid levels. The patterns were distinctly different from breast cancer or schizophrenia, as the strongest markers of lipid levels are of low/limited correlations with rs2965183/rs2905426 ($r^2 < 0.16$; Supplementary Fig. 4). Thus the signal at 19p13 shared by breast cancer and schizophrenia is unlikely explained by the genetic susceptibility to aberrant levels of lipids. We further searched other reported associations in this region using GWAS Catalog[28] and found 49 associations with 32 phenotypes (broadly categorized into lipids, blood count, psychobehavioral profiles, and others) in addition to breast cancer and schizophrenia (Supplementary Table 4). However, these associations, except for basophil count and red blood cell distribution width ($r^2$ from 0.813 to 0.871), are less correlated with rs2965183/rs2905426 ($r^2 < 0.6$). This suggests that the signal at 19p13 shared by breast cancer and schizophrenia is less likely explained by the genetic susceptibility to the aforementioned traits.

Next, we performed the *cis*-expression quantitative trait locus (eQTL) analysis of these two SNPs (rs2965183 and rs2905426) for genes up to 1 Mb on either side of each variant. We used gene expression data from tissues of breast (N = 251), brain (N = 80–154, depending on brain regions), or blood samples (N = 369) in the Genotype-Tissue Expression (GTEx) Project[29]. In blood samples, the risk alleles (A) of rs2965183 and (G) of rs2905426 were associated with elevated expression of nearby genes *GATAD2A* (corrected $P = 3.34 \times 10^{-15}$) and *TSSK6* (corrected $P = 4.15 \times 10^{-5}$) but reduced expression of *LPAR2* (corrected $P = 1.83 \times 10^{-18}$) and *MAU2* (corrected $P = 2.19 \times 10^{-4}$; Supplementary Table 5). Both

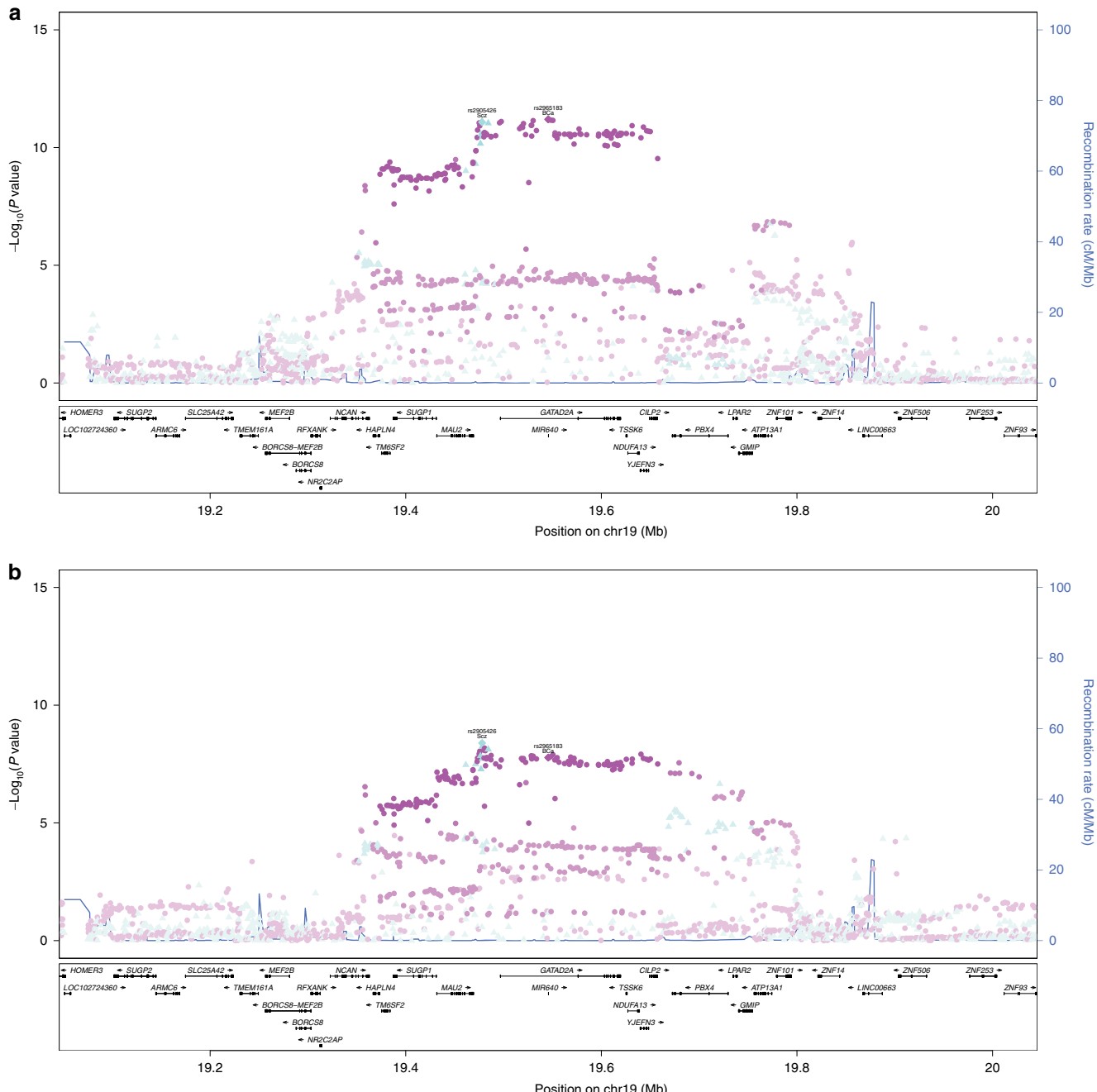

**Fig. 3 Regional association plot of GWAS summary statistics from breast cancer and schizophrenia for the locus 19p13. a** Associations with breast cancer. **b** Associations with schizophrenia. SNPs are colored corresponding to which of the lead SNPs it is in highest LD with, and the gradient of color denotes the extent of LD (i.e., lighter color means lower LD). This locus is anchored by lead SNPs rs2965183 (magenta) and rs2905426 (turquoise) for breast cancer and schizophrenia, respectively. GWAS genome-wide association study, LD linkage disequilibrium, SNP single-nucleotide polymorphisms.

variants were associated with increased level of *TM6SF2* in brain tissues (corrected *P* values <0.01). Moreover, the variants of rs2965183 and rs2905426 were related to higher expression of *HAPLN4* in breast (corrected $P = 0.001$) and brain tissues (corrected $P = 1.9 \times 10^{-4}$), respectively. All the aforementioned genes were nearby locus 19p13.

**Potential pathways shared by breast cancer and schizophrenia.** As evidenced by the PRS analysis and the significant genetic correlation between schizophrenia and breast cancer, it is unlikely that the association is only driven by the signal at 19p13. We,

therefore, performed a pathway analysis to identify more common mechanisms. We first mapped the overall genetic architecture shared by breast cancer and schizophrenia using cross-trait meta-analysis with adjustment for overlapping controls. To avoid detecting signals unduly driven by association from one side, we restricted the analysis to 946 SNPs of $P < 5 \times 10^{-5}$, of which 897 SNPs (94.8%) were with effect sizes in the same direction for both traits. These SNPs came from 17 genomic loci (Supplementary Fig. 5) and were mapped to 84 genes confirmed by either position mapping or eQTL analyses using breast, brain, or blood samples from GTEx. Utilizing databases including Gene Ontology (GO) Biological Process, BioCarta, Reactome, and

Kyoto Encyclopedia of Genes and Genomes (KEGG), we detected an enrichment of 176 pathways (all with $P < 0.05$ after correction of multiple comparison separately in each database, Supplementary Data 1). The most significantly enriched pathway regulates the biosynthesis of peptide hormones (e.g., corticotropin) that mediate neurotransmission and hormonal action (corrected $P = 1.57 \times 10^{-5}$).

## Discussion

To the best of our knowledge, this is the first in-depth dissection, from phenotypic to genetic associations, of the link between breast cancer and schizophrenia. Leveraging the Swedish nationwide population-based cohort, we showed an epidemiological bidirectional association between breast cancer and schizophrenia. Consistently, we further illustrated the genetic association of both traits using GWAS summary statistics from international consortia and identified a distinct locus (19p13) shared between breast cancer and schizophrenia using regional association plots. The noteworthy genetic overlap between breast cancer and schizophrenia may partly explain the phenotypic association found in the epidemiological analysis, and the identified shared locus may provide leads to the potential common biology for both traits.

Despite lower screening attendance[19], it has recently been recognized that patients with schizophrenia are at increased risk of breast cancer[6] but not of cancer overall[30,31]. Our results demonstrate that women with breast cancer are also at elevated risk of subsequent schizophrenia—largely representing a late-onset group documented as the second incidence peak following menopause[26]. It is plausible that many patients with schizophrenia who developed breast cancer later were early-onset ones, which might be phenotypically different from those developed after breast cancer. Our additional analysis restricting to women aged 18–44 years lends some support to a bidirectional association between early-onset schizophrenia and breast cancer, although the statistical power was very limited. The longitudinally bidirectional association in epidemiological analysis may point toward a partially shared etiology. Common behavioral or lifestyle factors (including nulliparity and obesity) have previously been suggested[6], but our results remained robust in an effort to control for these factors. It is also hypothesized that some treatment for schizophrenia, for instance, antipsychotics, may induce higher level of prolactin and subsequently increased risk of breast cancer[6]. Nevertheless, use of antipsychotics may not explain the excess risk of schizophrenia among patients with breast cancer. Altogether, these evidences lend some support to a potential genetic overlap between both diseases.

A recent study reporting shared genetic variants between schizophrenia and lung cancer highlighted a locus related to nicotinic acetylcholine receptors and smoking[32]. In the same study, based on a much smaller GWAS consortium of breast cancer (the Genetic Associations and Mechanisms in Oncology consortium (GAME-ON)), a weaker association between schizophrenia and breast cancer was detected by the stratified quantile–quantile plot and no further analysis was pursued[32]. Recently, both Byrne et al.[20] and Shi et al.[21] performed Mendelian randomization analyses based on similar GWAS summary statistics, which indicate that the genetic determinants of schizophrenia are associated with an increased risk of breast cancer. Another study based on GAME-ON data does, however, not support a causal genetic correlation between the two diseases[33]. Together with the longitudinal, bidirectional association observed in the epidemiological analysis and the weak pattern in the scatter plot of the lead SNPs, our findings also suggest a genetic overlap between the two traits rather than that one causing the other. We

estimated that around one-seventh of the genetic contribution to breast cancer is shared with genetic contribution to schizophrenia, which is in line with previous reports[20,34]. In fact, the genetic correlation of 0.14 between schizophrenia and breast cancer (as a somatic disease) is noteworthy in comparison with genetic correlation between some psychiatric disorders, for example, between schizophrenia and autism ($r_g = 0.14$)[35]. Of note, the genetic overlap is also noted between breast cancer and other psychiatric traits including depressive symptoms and neuroticism[34]. By contrast, some non-significant genetic correlations were detected between schizophrenia and other cancer types (e.g., ovarian cancer) but no correlation with colorectal cancer[34].

The novelty of our study further lies in the discovery of a distinct signal at 19p13 that is associated with both breast cancer and schizophrenia. A careful search of previously reported associations in this region support that this shared locus is not clearly explained by other traits. Importantly, this locus has also been identified as a shared locus associated with three hormone-related cancers, including breast, ovarian ($P = 0.0063$), and prostate cancer ($P = 0.01$)[36], although a precise biological pathway cannot be pinpointed by these data. The index cross-cancer variant rs1469713 is located between the lead SNPs of breast cancer (rs2965183, 16 kb downstream) and schizophrenia (rs2905426, 50 kb upstream). rs2965183 is located in enhancer elements reported to interact with GATAD2A in normal and tumor breast cells[36]. Our results also show that the locus 19p13 is associated with increased expression of GATAD2A in blood samples, while others observed a similar pattern in brain tissue[37,38]. GATAD2A is a subunit of the nucleosome remodeling and histone deacetylase (NuRD) complex. As a chromatin-level regulator of transcription, the NuRD complex may facilitate breast cancer development and progression by interacting with YAP/TAZ from the Hippo tumor-suppressor pathway[39], modulating estrogen functions and the human epidermal growth factor receptor 2 pathway[40], and by blocking p53-induced apoptosis[40]. Moreover, the locus 19p13 is associated with intellectual performance among broad psychosis cases[38]. GATAD2A is related to greater prefrontal activity[41], while NuRD complex regulates gene expression with temporal precision[42]. The results of several studies indicate that GATAD2A may be one of the causal genes in the pathogenesis of schizophrenia, potentially through dopamine receptor D2 and nerve growth factor (NGF) pathways[41,43,44]. NGF signaling was also enriched in our pathway analysis.

As indicated by the PRS analysis and cross-trait meta-analysis, other weakly associated variants besides the signal at 19p13 may also pillar the shared genetic architectures between breast cancer and schizophrenia. In the pathway analysis accounting for weaker signals, we identified several pathways that are potentially underlying the two traits. The most significant one is the peptide hormone biosynthesis pathway (corrected $P = 1.57 \times 10^{-5}$), which regulates the processing of corticotropin (known as adrenocorticotropic hormone (ACTH)), a well-known player in hypothalamic–pituitary–adrenal axis and hence the stress response. ACTH and other stress hormones are closely related to many mental disorders, including schizophrenia[45,46], and have a pivotal role in cancer development, potentially interacting through immune system and sex hormones[47,48]. The genetic association between schizophrenia and ER-positive breast cancer indeed appeared slightly stronger in our data, although it could be partially explained by the larger sample size of ER-positive breast cancer.

One major strength of the present study is the use of large-scale population data, with high validity data on the diagnoses of breast cancer and schizophrenia[49,50], and genetic data from multiple sources, to illustrate the consistent association of breast cancer with schizophrenia risk and vice versa. Nevertheless, we

acknowledge several limitations. Despite the efforts of addressing behavior-related factors (e.g., smoking-related comorbidities and obesity by using proxies of clinically diagnosed tobacco use disorder and obesity, respectively), the residual confounding cannot be ruled out. For instance, we have no information whether the obesity was pre- or post-menopausal. However, many individuals had first episode of schizophrenia in their 20s/30s and were subsequently at risk for overweight/obesity due to antipsychotic treatment[51,52] yielding decreased risk of breast cancer[53], which suggests that our estimates are rather conservative. We also lack information on hormone levels, e.g., estradiol. However, women with schizophrenia often have a lower level of circulating estradiol[54], which would have led to decreased risk of breast cancer[55,56]. Moreover, some treatments for schizophrenia (e.g., antipsychotics) may induce higher levels of prolactin and thereby increasing the risk of breast cancer[6], whereas breast cancer treatment (e.g., hormone therapy[57] and chemotherapy[58]) may effect on brain functions. It is plausible that genetic overlap, together with potential behavioral factors and treatment, contribute to the observed phenotypic association between breast cancer and schizophrenia. Replication of the PRS analysis in individually genotyped studies independent of these two international consortia would have been preferable. Nevertheless, the genetic association between breast cancer and schizophrenia has been observed in several studies using different data sources[32,33]. In addition, although we showed that the locus 19p13 shared by breast cancer and schizophrenia is independent of a number of traits including blood lipid levels, an association of this locus with ovarian and prostate cancer has also been suggested[36]. Thus future studies are needed to understand the potential genetic overlap between schizophrenia and other hormone-related cancers as well as the link between schizophrenia and male breast cancer.

In conclusion, our findings suggest that breast cancer and schizophrenia are both phenotypically and genetically correlated. The noteworthy genetic contribution to both traits, highlighted by a shared locus at 19p13 and among other loci, has the potential to shed light on the common biology underlying breast cancer and schizophrenia. Functional studies are needed to uncover how the identified genetic signals contribute to the pathogenesis of both diseases.

## Methods

**Data sources.** Based on the Swedish Population and Housing Census in 1990, we identified all women born and living in Sweden in 1990 ($N = 3,937,114$). Using the unique national identification numbers, we followed all women through cross-linkages to the Swedish Cancer Register and Patient Register to identify occurring cases of breast cancer and schizophrenia, respectively. The Cancer Register has been available since 1958 and approaches 100% completeness. The Patient Register has information on ≥80% of hospital discharge records since 1980s, complete records since 1987, and ≥80% of hospital-based outpatient visits since 2001. The Cause of Death and Migration Registers were also cross-linked for follow-up. Demographic characteristics were obtained from the Census. Information on educational level was retrieved from the Education Register, while parity at matching was obtained from the Medical Birth Register and the Total Population Register. In addition, diagnoses of other psychiatric disorders and obesity were identified from the Patient Register. See Supplementary Table 6 for all diagnostic codes that were used.

The multi-stage GWAS of breast cancer was recently conducted among 122,977 cases and 105,974 controls of women of European ancestry from 68 studies collected by the Breast Cancer Association Consortium and Discovery, Biology and Risk of Inherited Variants in Breast Cancer Consortium[23]. In each of the participating studies, genotype data were imputed using the 1000 Genomes Project reference panel (1KG; Phase 3) and filtered for a minor allele frequency (MAF) ≥ 0.5% and imputation quality score ≥0.3, retaining 11.8 million variants. Association testing was conducted on these imputed markers with adjustment for countries and ancestry-informative principle components (PCs). A fixed-effect meta-analysis was conducted combining each result and results from the Collaborative Oncological Gene–environment Study and other 11 independent GWASs of breast cancer.

The multi-stage GWAS of schizophrenia were performed on up to 36,989 cases and 113,075 controls by the Schizophrenia Working Group of the Psychiatric

Genomics Consortium (PGC)[24]. We used the summary statistics from 33,640 cases and 43,456 controls of European ancestry from 46 ancestry matched, non-overlapping case–control samples and 3 family-based samples (public data downloaded from the PGC website). Unified quality control (QC) procedures, Ricopilli pipeline, were used and genotype data were imputed using the 1KG (Phase I Integrated Release Version 3). In each sample, associations were assessed for imputed marker dosages by adjusting for PCs to control for population stratification. After filtering on MAF ≥ 0.01, imputation quality score ≥0.6, about 9.5 million variants were used for a combined inverse-variance weighted fixed-effect analysis.

To harmonize the GWAS summary statistics of breast cancer and schizophrenia, we filtered the imputed markers by MAF ≥ 0.01, imputation quality score ≥0.6 ($R^2$ from MACH[59] or INFO from IMPUTE[60]), removing duplicates, insertion/deletion, and allele-mismatched variants. Six million seven hundred and twenty-five thousand eight hundred and eighteen SNPs in autosome shared between two GWAS were included for analysis.

In order to disentangle the potential confounding of genetic susceptibility to lipid levels, we also utilized the GWAS summary statistics of TC, TG, and LDL cholesterol as a negative control. We obtained the GWAS summary statistics of blood lipids from the Global Lipids Genetics Consortium[61], which included about 100,000 individuals with European ancestry ($N_{TC} = 100,184$, $N_{TG} = 96,598$, and $N_{LDL} = 95,454$) and 2.69 million SNPs. To be consistent, we only used SNPs that were reported in GWASs of breast cancer and schizophrenia.

The individual-genotype data for schizophrenia was based on the S3 as part of the Schizophrenia Working Group of PGC[24]. Cases of this study were adults with two or more hospitalization due to schizophrenia or schizoaffective disorder, ascertained through the Swedish Patient Register. Controls were randomly selected from the Swedish population and had never been hospitalized for schizophrenia or bipolar disorder. The data genotyped on different arrays were processed using the PGC Ricopilli pipeline for QC and imputation[24].

The register-based study was approved by the Central Ethical Review Board in Stockholm, Sweden (Dnr 12–2013). The requirement of informed consent is waived in register-based studies in Sweden. The S3 were approved by the Regional Ethical Review Board in Stockholm, Sweden (Dnr 04/449/4), and informed consent was obtained from every participant. Data from international consortia (summary statistics) are publicly available.

**Epidemiologic analyses.** First, we examined schizophrenia-associated risk of invasive breast cancer using a nested case–control design. Among 3,937,114 women, we excluded 39,990 women with breast cancer, 81,692 with other malignancy, and 48,320 with emigration before January 1, 1990, or age 18 years, whichever came later, and 982 women with erroneous records. We followed the remaining 3,766,130 women until the diagnosis of any cancer, death, emigration, December 31, 2009, whichever came first. The average duration of follow-up was 15.2 years.

To assess the schizophrenia-associated risk of breast cancer, we first conducted a nested case–control analysis, which preserves the validity of full prospective cohort analysis[22]. Briefly, within this nationwide study base, we identified 94,626 women with newly diagnosed invasive breast cancer as cases. For each case, 30 women (whenever possible) who were free of cancer at the date of the case's cancer diagnosis (i.e., the reference date) were randomly selected from the study base. In total, 2,838,765 controls were individually matched on year of birth and age at the reference date. We then ascertained the date of a first-ever inpatient diagnosis of schizophrenia before the reference date (since 1980). The median time from schizophrenia to breast cancer diagnosis was 12.74 years. Both primary and secondary diagnoses were used. To capture persons with schizophrenia who might not have been hospitalized, we performed an additional analysis including outpatient diagnoses of schizophrenia.

We derived ORs and 95% CIs from the conditional logistic regression by comparing cases with controls. The estimates, however, should be interchangeably interpreted as relative risk of breast cancer among patients with schizophrenia[22]. The estimates were conditioned on the matching sets (i.e., inherently adjusted for birth of year and age) and adjusted for educational levels (primary school, high school, college and beyond, or unknown) and region of residence (southern, central, or northern Sweden). To disentangle the environmental effect from potential genetic factors, in a second model, we further adjusted for characteristics shared between breast cancer and schizophrenia, including parity (0, 1–2, or ≥3), pre-existing psychiatric disorder (yes or no; including all substance use disorders, e.g., abuse of alcohol, tobacco, and drug etc.), and obesity (yes or no) at the reference date.

Second, we assessed invasive breast cancer-associated risk of schizophrenia using a matched cohort design. We prospectively followed these cases and controls from the reference date until diagnosis of schizophrenia, cancer, death, emigration, December 31, 2010, whichever came first. We excluded 548 cases and 11,399 controls with inpatient or outpatient diagnosis of schizophrenia before the reference date. During the follow-up (i.e., from the reference date onward; median, 7.72 years), we further ascertained newly diagnosed schizophrenia in inpatient (and outpatient) care. We calculated the hazard ratio of schizophrenia and 95% CI by using the stratified Cox proportional hazards regression comparing cases and controls. The approaches of adjustment in models were the same as the nested case–control study.

In an additional analysis, we restricted the analysis to women of age at follow-up during 18–44 years. Namely, we included women whose age at the reference date was 18–44 years for both analysis; in the analysis of breast cancer-associated schizophrenia risk, we censored the follow-up when the individual turned age 45 years.

These analyses were conducted in STATA (version 14.2). All tests were two sided and $P < 0.05$ indicated statistical significance.

**PRS analyses**. First, we tested the association between PRS for schizophrenia and risk of breast cancer and vice versa, using the GWAS summary statistics from breast cancer and schizophrenia. The SNPs reported in both GWAS were included to generate the PRS using the software packages PRSice (version 1.25)[62]. LD clumping ($r^2 < 0.1$ in 500 kb window) used 1000 Genomes Project European samples as LD reference (after removing MHC region). Similarly, subgroup analyses on ER-positive/negative breast cancer were also performed.

Second, to examine the association of breast cancer with schizophrenia in women and men, we derived PRS for breast cancer for all participants in S3. The harmonized GWAS summary statistics of breast cancer was considered as discovery set for SNP selection and risk allele weighting. We further excluded variants with MAF < 0.05, imputation quality score <0.8, or with ambiguous strand. After LD clumping ($r^2 < 0.1$ in 500 kb window) using the same LD reference as above, 71,121 SNPs with relative independence remained for PRS calculation. We generated PRS for breast cancer in the target sample as the sum of the imputed SNP dosages weighted by the allele effect (logistic regression coefficients) from the discovery set across all SNPs under a $P_T$ of $5 \times 10^{-8}$, $1 \times 10^{-6}$, $1 \times 10^{-4}$, $1 \times 10^{-3}$, 0.01, 0.05, 0.1, 0.2, 0.5, and 1, respectively. PLINK (version 1.9) was used for the PRS profiling. All scores were standardized within each target set to account for variation in SNP numbers used for PRS calculation. Subsequently, we examined the associations using logistic regression in R (version 3.4.3). The variance explained by PRS was calculated as the difference in variance from the full model including the PRS and variance from the baseline model without PRS using Nagelkerke's $R$-squared[63]. We adjusted for first ten PCs and genotyping platform.

**LD score genetic correlation analyses**. We estimated the genetic correlation, i.e., the extent to which breast cancer and schizophrenia share genetic etiology, by using LD score correlation[64]. This method requires only GWAS summary statistics and the estimation is always controlled for sample overlap (if any). It is implemented in the free open source LDSC (version 1.0.1) software package. Briefly, this method quantifies the separate contributions of polygenic effects by examining the relationship between LD score and test statistics of SNPs from GWAS summary statistics and produces genetic correlation based on the deviation of *chi-square* statistics from the null hypothesis. The genetic correlation was estimated by using unconstrained-intercept LD score regression accounting for potential sample overlap between GWASs of breast cancer and schizophrenia.

**Identification of shared loci**. To identify the most significant loci shared between breast cancer and schizophrenia, we first plotted the associations of schizophrenia lead SNPs (i.e., genome-wide significant markers after LD clumping based on $r^2 < 0.1$ in 500 kb window) with breast cancer, by using the GWAS summary statistics of both traits. A global test that approximates the average effects of a set of selected SNPs on the other trait was carried out to highlight the overall effect of lead SNPs for schizophrenia on breast cancer[65]. Moreover, the individual effect of lead SNPs for schizophrenia on breast cancer was corrected for multiple comparison using the false discovery rate method[66]. The remained significant SNPs, regardless of effect direction, stand as potential regions shared with breast cancer. In addition, to assess the effect of influential SNPs, we calculated the Cook's distance for each SNP and those with Cook's distance >$N/4$ were removed. We further plotted the association by removing SNPs iteratively until the heterogeneity test was no longer significant. This procedure was done by using *grs.filter.Qrs* function in the *gtx* package in R. Identical analyses were performed for breast cancer lead SNPs and their corresponding associations with schizophrenia.

Next, we visualized the regional associations (500 kb on either side of the lead SNPs that remained significant after correction for multiple comparison) with both breast cancer and schizophrenia by integrating 1000 Genomes LD data with gene annotation tracks using LocusZoom (version 1.4)[67]. As the signal at 19p13 shows the similar pattern between both traits, we sought the known associations of lead SNPs rs2965183 and rs2905426 in PhenoScanner database (http://www.phenoscanner.medschl.cam.ac.uk/)[27]. Based on the established links of both SNPs to lipids, we further plot the regional associations of locus 19p13 with TC, TG, and LDL using the GWAS summary statistics. We also sought reported associations of this region from GWAS Catalog (https://www.ebi.ac.uk/gwas/home) and calculated $R^2$ to rs2965183/rs2905426 in European population using LDmatrix (https://ldlink.nci.nih.gov/?tab=ldmatrix).

**Gene expression analyses**. To capture potentially altered gene expression due to lead SNPs rs2965183 and rs2905426, the eQTL analyses were then performed for genes up to 1 Mb on either side of each variant. Data on genetic variant and expression in breast ($N = 251$), brain ($N = 80–154$ depending on brain regions), or blood samples ($N = 369$) were used, through FUMA, from the GTEx Project[29].

**Cross-trait meta-analysis of GWAS summary statistics**. To better capture the genetic architecture shared by both traits, we performed cross-trait meta-analysis by combining GWAS summary statistics for breast cancer and schizophrenia. We applied metaUSAT (implemented in the R software package), a data-adaptive statistical approach for multivariate meta-analysis[68]. In brief, metaUSAT can be applied to test the genetic components of multiple traits by using only univariate summary statistics ($Z$-score) from multiple studies. It provides an approximate asymptotic $P$ value for association and appropriately accounts for sample overlap between studies by estimating the correlation matrix of summary statistics. To avoid detecting signals unduly driven by association from one side, we only included SNPs with $P$ value $<5 \times 10^{-5}$ reported in both GWAS summary statistics for meta-analyses.

**Functional mapping and annotation**. We used FUMA[69], a web-based platform (http://fuma.ctglab.nl/), to perform functional mapping and annotation for the shared genetic components associated with both breast cancer and schizophrenia. To functionally annotate GWAS findings, FUMA uses information from 18 biological data repositories and tools to run two core processes. Based on the uploaded results of cross-trait meta-analysis, 32 independent significant SNPs were identified by FUMA (LD $r^2 < 0.6$, $P_{meta} < 1 \times 10^{-5}$), and 84 genes were mapped in either positional (i.e., SNPs physically located inside a gene with up to 10 kb windows) or eQTL mapping (based on breast, brain, or blood samples from the GTEx project as described above). Pathways were then enriched from these genes using knowledge-based databases, including Reactome, KEGG, GO Biological Process, and BioCarta. The $P$ values in eQTL and pathway analyses were corrected for multiple comparison using the false discovery rate method[66].

**Reporting summary**. Further information on research design is available in the Nature Research Reporting Summary linked to this article.

## Data availability

The epidemiological analysis used Swedish registers are available from the Swedish National Board of Health and Welfare (https://www.socialstyrelsen.se/en/statistics-and-data/registers/) and the Statistics Sweden (https://www.statistikdatabasen.scb.se/pxweb/en/ssd/). According to Swedish law, the authors are not able to make the dataset publicly available. Data are, however, available for any researchers (including international researchers) through formal applications to the aforementioned authorities. The primary genetic analysis used publicly available GWAS summary statistics from the Breast Cancer Association Consortium (available at http://bcac.ccge.medschl.cam.ac.uk/bcacdata/icogs-complete-summary-results/) and Psychiatric Genomics Consortium (available at https://www.med.unc.edu/pgc/download-results/). We also used reported genetic associations from PhenoScanner database (http://www.phenoscanner.medschl.cam.ac.uk/) and GWAS Catalog (https://www.ebi.ac.uk/gwas/home). Source data are provided with this paper.

## Code availability

All code used for data preparation and primary analysis are deposited on GitHub.

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

## Acknowledgements

This work is supported by the Swedish Research Council (grant number: 2018-00648, to D.L.), Karolinska Institutet Research Foundation (grant number: 2018-01585, to D.L.),

Grant of Excellence, Icelandic Research Fund (grant number 163362-051, to U.A.V.), and ERC Consolidator Grant (StressGene, grant number:726413, to U.A.V.). We thank Dr. Agnar Helgason, Dr. Patrick Sulem, and Dr. Kári Stefánsson from deCODE Genetics, Iceland for helpful discussions on data analysis and interpretations. We also acknowledge the shared GWAS statistical summaries of schizophrenia from the Psychiatric Genetics Consortium and breast cancer from the Breast Cancer Association Consortium.

## Author contributions

D.L. and U.A.V. conceived this study, while D.L., J.S., R.M.T., and U.A.V. contributed to the study design. D.L. and J.S. performed the statistical analyses. P.S. provided the data access to the Swedish Schizophrenia Study. D.L., J.S., and U.A.V. drafted the manuscript. All authors contributed to the data interpretation and manuscript writing and approved the final draft. R.M.T. and U.A.V. shared the supervision and administration.

## Funding

## Competing interests

The authors declare no competing interests.
