## [Peer Review File · Nature Communications]

Reviewers' comments:

Reviewer #1 (Remarks to the Author):

The manuscript, A Shared Genetic Contribution to Breast Cancer and Schizophrenia, using findings of GWAS and population-level data, proposes a bidirectional association of breast cancer and schizophrenia risk. The findings are supported in part by prior studies suggesting a link between the two conditions and raise a number of interesting questions about the roles of genetic susceptibility, stress, hormones and other factors. The findings are novel and would be of interest to a broad range of providers and researchers.

The potential bi-directionality is new and intriguing and raises questions in particular about the patients with breast cancer who go on to develop schizophrenia. These persons represent a considerably smaller cohort (100 vs. 534 hospitalized for schizophrenia diagnosis). There is an obvious question about diagnostic reliability, though details would not likely be knowable. Risk for breast cancer increases with age. The finding of risk of schizophrenia after breast cancer raises the possibility of a (psychiatric) phenotypic difference from those patients who started with schizophrenia and later developed breast cancer, namely paraphrenia or late-onset schizophrenia, as is mentioned, which can run a very different life-course and be qualitatively different from the earlier-onset form. It is also more common in women. Were the patients in the second group older at age of schizophrenia diagnosis compared to the other cohort? It would be interesting to know as well any differences in time lag between diagnoses. A phenotypic distinction might also explain why the people with breast cancer then later schizophrenia had proportionally greater (2.6 vs. 26%) outpatient diagnoses.

The limitations section does mention some behavior-related factors and there is prior acknowledgement that unmeasured confounders may be involved. Other exposure-related risk factors for breast cancer including hormone levels, for example, don't appear to be accounted for. Did the models include data on alcohol exposure or alcohol use disorder (or other substance) diagnoses? Did the second model adjust solely for obesity or post-/peri-menopausal obesity? The latter is the known risk factor. Premenopausal obese women are thought to have a lower risk for breast CA.

The study appears to include only women, though men's risk might increase as well despite overall lower risk for breast cancer.

Reviewer #2 (Remarks to the Author):

In this manuscript, Lu et al. investigate the relationship between schizophrenia and breast cancer using a comprehensive range of observational and genetic epidemiological large-scale data sets and state-of-the-art methods.

1. The conclusion of bi-directionality of the *genetic* relationship between schizophrenia and breast cancer hinges almost entirely on the polygenic risk score (PRS) analyses. The genetic correlation, identification of shared genetic risk loci, and pathway-level genetic analyses do not provide much information with respect to direction. Further, the conclusions in this manuscript contradict previous Mendelian randomization studies based on similar data sets (those manuscripts, perhaps correctly, conclude that the relationship is unidirectional). This leads to the following major concerns about conclusions drawn on the basis of PRS analyses:

- Figure 2 A indicates that the association between a PRS based on the lead SNPs for schizophrenia and breast cancer is remarkably heterogeneous. This would suggest (as suggested by the figure too) that only some schizophrenia lead SNPs drive the association. It would be helpful to identify influential SNPs in the genetic instrument using standard regression diagnostic tools, such as calculating Cook's distances and/or Studentized residuals for all the SNPs/datapoints and

performing leave one out and leave all influential points out analyses. How is the association affected if SNPs are iteratively removed from the risk score until the heterogeneity test is no longer significant at the specified threshold? This can be performed using the `grs.filter.Qrs` function in the R package "gtx" genetics toolbox (<https://rdrr.io/cran/gtx/man/grs.filter.Qrs.html>). Likewise, how do these additional analyses affect the association between breast cancer and schizophrenia shown in Figure 2 B (which is also heterogeneous)? What can we learn from the SNPs contributing to the heterogeneity (see for example the tool MR-TRYX from Cho, Hemani, et al)?

- Figure 1 B indicates that the association between the breast cancer PRS and schizophrenia only really appears when large numbers of SNPs with weak breast cancer associations or even SNPs not associated with breast cancer are included in the breast cancer PRS. This seems very strange and incompatible with a "bi-directional" relationship. Can the authors perform additional analyses to prove that this is not an artefact? Even the observational estimates for association of breast cancer with subsequent schizophrenia in Table 1 are quite weak this taken together with the genetics further weakens the case for bi-directionality. How would Figure 1 B be affected by control analyses (say prostate/ovarian cancer PRS -> schizophrenia and breast cancer PRS -> psychiatric disease other than schizophrenia – all these data sets are publicly available)?

2. Given the use of genetic correlations and PRSes in this manuscript, the latent causal variable method of O'Connor and Price (PMID: 30374074) may further inform the dissection of the relationship between schizophrenia and breast cancer.

Reviewers' comments:

Reviewer #1 (Remarks to the Author):

The manuscript, A Shared Genetic Contribution to Breast Cancer and Schizophrenia, using findings of GWAS and population-level data, proposes a bidirectional association of breast cancer and schizophrenia risk. The findings are supported in part by prior studies suggesting a link between the two conditions and raise a number of interesting questions about the roles of genetic susceptibility, stress, hormones and other factors. The findings are novel and would be of interest to a broad range of providers and researchers.

Response: Thanks for the positive comments.

The potential bi-directionality is new and intriguing and raises questions in particular about the patients with breast cancer who go on to develop schizophrenia. These persons represent a considerably smaller cohort (100 vs. 534 hospitalized for schizophrenia diagnosis). There is an obvious question about diagnostic reliability, though details would not likely be knowable.

Response: Thanks for raising this issue. The validity of the schizophrenia diagnosis is considered high in the Swedish Patient Register with a positive predictive value of 0.94 (*Ekholm B, Nord J Psychiatry, 2005, 59:457-64*). We have now added this to the Discussion.

Modified text:

Page 15, paragraph 2: One major strength of the present study is the use of large-scale population data, with high validity data on the diagnoses of breast cancer and schizophrenia^{49,50}, and genetic data from multiple sources, to illustrate the consistent association of breast cancer with schizophrenia risk and vice versa.

There are two major reasons that the number of schizophrenia diagnoses is larger before breast cancer diagnosis than that after breast cancer diagnosis. First, most schizophrenia diagnoses occur in the late 20s to early 30s in women, which contributes to the number of schizophrenia diagnoses before breast cancer diagnosis. On the other hand, the late-onset schizophrenia, as acknowledged in our Discussion and the reviewer's comment below, contributes primarily to the number of schizophrenia diagnoses after breast cancer diagnosis and represents a relatively small proportion of all women with schizophrenia. Second, we tracked any schizophrenia diagnosis before breast cancer (diagnosed during 1990-2009) back to 1980 and the median time from schizophrenia to breast cancer diagnosis was 12.74 years. By contrast, we only followed the patients after breast cancer diagnosis for a first-ever diagnosis of schizophrenia until 2010 and the median follow-up is much shorter (7.72 years). We have now added this point to the Methods.

Modified text:

Page 20, paragraph 2: We then ascertained the date of a first-ever inpatient diagnosis of schizophrenia before the reference date (since 1980). The median time from schizophrenia to breast cancer diagnosis was 12.74 years.

Page 21, paragraph 2: During the follow-up (i.e., from the reference date onward; median, 7.72 years), we further ascertained newly diagnosed schizophrenia in inpatient (and outpatient) care.

Risk for breast cancer increases with age. The finding of risk of schizophrenia after breast cancer raises the possibility of a (psychiatric) phenotypic difference from those patients who started with schizophrenia and later developed breast cancer, namely paraphrenia or late-onset schizophrenia, as is mentioned, which can run a very different life-course and be qualitatively different from the earlier-onset form. It is also more common in women. Were the patients in the second group older at age of schizophrenia diagnosis compared to the other cohort? It would be interesting to know as well any differences in time lag between diagnoses. A phenotypic distinction might also explain why the people with breast cancer then later schizophrenia had proportionally greater (2.6 vs. 26%) outpatient diagnoses.

Response: We agree with the reviewer that there are more early-onset schizophrenia cases among the cases ascertained before breast cancer diagnosis, whereas more late-onset cases among all cases of schizophrenia ascertained after breast cancer diagnosis. In the analysis of schizophrenia-associated breast cancer risk, the mean age at schizophrenia diagnosis was 48.9 years (SD, 14.8 years), which is the first diagnosis since 1980 and likely not the age at first episode for schizophrenia. In the analysis of breast cancer-associated schizophrenia risk, the mean age of schizophrenia diagnosis is 67.8 years (SD, 12.7 years).

Regarding the interval time between diagnoses, the elapsed time from schizophrenia to breast cancer is 12.74 years as mentioned in the response to Comment #1. In the analysis of schizophrenia risk after breast cancer, the median time from breast cancer to schizophrenia is 5.03 years.

The small increase by adding outpatient diagnoses in the former analysis is because, when tracking schizophrenia diagnoses up to 29 years (during 1980-2009) before breast cancer, patients with schizophrenia were more likely to be identified by at least one hospitalization for schizophrenia during this long-period. Only individuals never made a hospitalization for schizophrenia during the time would be captured by adding the outpatient diagnoses. As mentioned above, the follow-up for the analysis of schizophrenia risk after breast cancer was shorter and thus there were more patients identified by adding the outpatient diagnoses.

To address the concern of phenotypic distinction between groups, we have now performed an additional analysis restricted to women aged 18-44 years at follow-up (Table S1 below) given that late-onset schizophrenia is hypothesized to be related to menopause. This analysis further supports the bidirectional association between early-onset schizophrenia and breast cancer, although the numbers of schizophrenia are small and estimates are not statistically significant. We have now added this analysis to the manuscript and expanded the discussion on early-onset vs. late-onset schizophrenia in the Discussion.

Modified text:

Page 6, paragraph 3: To shed light on the early-onset schizophrenia, we restricted the analysis to women aged 18-44 years and suggested a bidirectional, positive association between early-

onset schizophrenia and breast cancer, although the numbers were small and estimates were not statistically significant (Supplementary Table S1).

Page 12, paragraph 2: It is plausible that many patients with schizophrenia who developed breast cancer later were early-onset ones, which might be phenotypically different from those developed after breast cancer. Our additional analysis restricting to women aged 18-44 years lends some support to a bidirectional association between early-onset schizophrenia and breast cancer, although the statistical power was very limited.

Page 21, paragraph 3: In an additional analysis, we restricted the analysis to women of age at follow-up during 18-44 years. Namely, we included women whose age at the reference date was 18-44 years for both analysis; in the analysis of breast cancer-associated schizophrenia risk, we censored the follow-up when the individual turned age 45 years.

Table S1. Bidirectional association between invasive breast cancer and early-onset schizophrenia among women of age at follow-up during 18-44 years.

	Women without breast cancer	Women with breast cancer				
	N (%)	N (%)	OR (95% CI) [‡]	P [‡]	OR (95% CI) [§]	P [‡]
Association of schizophrenia with subsequent breast cancer*						
Number of women	258,450	8,615	-	-	-	-
Inpatient diagnosis of schizophrenia	900 (0.35)	32 (0.37)	1.10 (0.77-1.56)	0.610	1.08 (0.76-1.55)	0.666
	N (%)	N (%)	HR (95% CI) [‡]	P [‡]	HR (95% CI) [§]	P [‡]
Association of breast cancer with subsequent schizophrenia[†]						
Number of women	257,550	8,583	-	-	-	-
Inpatient diagnosis of schizophrenia	248 (0.10)	12 (0.14)	1.78 (0.98-3.21)	0.057	1.52 (0.69-3.38)	0.302

N, number; OR, odds ratio; HR, hazard ratio.

* Based on the nested case-control study. The estimates, i.e., OR derived from conditional logistic regression, should be interchangeably interpreted as the risk of breast cancer among patients with schizophrenia.

[†] Based on the matched cohort study. The estimates, i.e., HR derived from stratified Cox proportional hazards regression, are interpreted as the risk of schizophrenia among patients with breast cancer.

[‡] Models were adjusted for educational level (primary school, high school, college and beyond, or unknown), and region of residence (southern, central, or northern Sweden). Birth year and age at reference were inherently controlled for due to matching.

[§] Models were additionally adjusted for parity (0, 1-2, or ≥3), pre-existing psychiatric disorder (yes or no; including substance use disorders), and obesity (yes or no) at matching.

The limitations section does mention some behavior-related factors and there is prior acknowledgement that unmeasured confounders may be involved. Other exposure-related risk factors for breast cancer including hormone levels, for example, don't appear to be accounted for.

Response: That is a valid point. We do not have blood samples to measure the estradiol level, which is associated with increased risk of breast cancer. However, women with schizophrenia often have lower levels of circulating estradiol, which would have led to decreased risk of breast cancer. Accordingly, our estimates are rather conservative. We have now added this limitation in the Discussion.

Modified text:

Page 15, paragraph 2: We also lack information on hormone levels, e.g., estradiol. However, women with schizophrenia often have a lower level of circulating estradiol⁵⁴, which would have led to decreased risk of breast cancer^{55,56}.

Did the models include data on alcohol exposure or alcohol use disorder (or other substance) diagnoses?

Response: The information on alcohol consumption is not available in the registers. In our models, we however have already included diagnoses of substance use disorders (including abuse of alcohol, drug, and tobacco etc.) as recorded in the Patient Register. We have now added this information to the Methods.

Modified text:

Page 21, paragraph 1: To disentangle the environmental effect from potential genetic factors, in a second model, we further adjusted for characteristics shared between breast cancer and schizophrenia, including parity (0, 1-2, or ≥ 3), pre-existing psychiatric disorder (yes or no; including all substance use disorders, e.g., abuse of alcohol, tobacco, and drug etc.), and obesity (yes or no) at the reference date.

Did the second model adjust solely for obesity or post-/peri-menopausal obesity? The latter is the known risk factor. Premenopausal obese women are thought to have a lower risk for breast CA.

Response: Thanks for pointing this out. Since we lacked information on the timing of menopause, we have only adjusted for obesity which could be either pre- or post-menopausal. However, many were diagnosed with schizophrenia in their 20s-30s and were often overweight due to antipsychotic treatment (i.e., premenopausal overweight/obesity), which would have led to decreased risk of breast cancer. That said, our estimates are rather conservative. We have now added this to the Discussion.

Modified text:

Page 15, paragraph 2: Despite the efforts of addressing behavior-related factors (e.g., smoking-related comorbidities and obesity by using proxies of clinically diagnosed tobacco use disorder and obesity, respectively), the residual confounding cannot be ruled out. For instance, we have no information whether the obesity was pre- or post-menopausal. However, many individuals had first episode of schizophrenia in their 20s/30s and were subsequently at risk for overweight/obesity due to antipsychotic treatment^{51,52}, yielding decreased risk of breast cancer⁵³, which suggests that our estimates are rather conservative.

The study appears to include only women, though men's risk might increase as well despite overall lower risk for breast cancer.

Response: We agree with the reviewer that it is a very interesting question. As the number of male breast cancer is small in our study base, the statistical power would be very limited to analyze the association with schizophrenia, which is also not a common outcome. In addition, the genetic markers for male breast cancer are less established. We have now acknowledged this limitation in the Discussion.

Modified text:

Page 16, paragraph 1: *Thus, future studies are needed to understand the potential genetic overlap between schizophrenia and other hormone-related cancers as well as the link between schizophrenia and male breast cancer.*

Reviewer #2 (Remarks to the Author):

In this manuscript, Lu et al. investigate the relationship between schizophrenia and breast cancer using a comprehensive range of observational and genetic epidemiological large-scale data sets and state-of-the-art methods.

1. The conclusion of bi-directionality of the *genetic* relationship between schizophrenia and breast cancer hinges almost entirely on the polygenic risk score (PRS) analyses. The genetic correlation, identification of shared genetic risk loci, and pathway-level genetic analyses do not provide much information with respect to direction. Further, the conclusions in this manuscript contradict previous Mendelian randomization studies based on similar data sets (those manuscripts, perhaps correctly, conclude that the relationship is unidirectional). This leads to the following major concerns about conclusions drawn on the basis of PRS analyses:

Response: We understand that the term “*bi-directionality*” may be confused with the direction of causality using approaches such as Mendelian Randomization. In the original text, we have used this term to refer the longitudinal, bidirectional epidemiologic association, which is unlikely explained by causation but rather shared risk factors (*Rothman KJ, et al, Modern Epidemiology, third edition, 2008, pp 185*), such as shared genetic factors. Our genetic analysis has therefore focused on identifying genetic overlap between schizophrenia and breast cancer, which has no inference on causation as the Reviewer pointed out. In the analysis of lead SNPs (Figure 2), we also stated in the Results “When focusing on the 140 lead SNPs for breast cancer, we observed no overall correlation with schizophrenia ($P=0.425$).” This is in fact in line with previous findings which only observed a genetic relationship from schizophrenia to breast cancer (*Byrne EM, Schizophr Bull, 2019. 45(6):1251-1256. Shi J, J Psychiatr Brain Sci, 2018. 3(4)*). We have now clarified on the rational in the manuscript.

Modified text:

Page 7, paragraph 1: *Given the observation in the epidemiological analysis, we subsequently focused on confirming and identifying genetic overlap between schizophrenia and breast cancer in the genetic analysis. First, we calculated the genetic correlation between breast cancer and schizophrenia using linkage disequilibrium (LD) score regression.*

We have always tried to restrict our claims on the bidirectional association to the epidemiological analysis. For example, “Together with the longitudinal, bidirectional association observed in the epidemiological analysis and the weak pattern in the scatter plot of the lead SNPs, our findings also suggest a genetic overlap between the two traits rather than that one causing the other” (Page 13, paragraph 1). We have now throughout the manuscript clarified that claims on bidirectionality are limited to our findings from the epidemiological analysis and are not indicated in the genetic relationship.

Modified text:

Page 3, paragraph 1: *The epidemiological bidirectional association between breast cancer and schizophrenia may partly be explained by the genetic overlap between the two phenotypes and, hence, shared biological mechanisms.*

Page 12, paragraph 1: *Leveraging the Swedish nationwide population-based cohort, we showed an epidemiological bidirectional association between breast cancer and schizophrenia.*

Page 12, paragraph 2: *The longitudinally bidirectional association in the epidemiological analysis may point towards a partially shared etiology.*

- Figure 2 A indicates that the association between a PRS based on the lead SNPs for schizophrenia and breast cancer is remarkably heterogeneous. This would suggest (as suggested by the figure too) that only some schizophrenia lead SNPs drive the association. It would be helpful to identify influential SNPs in the genetic instrument using standard regression diagnostic tools, such as calculating Cook's distances and/or Studentized residuals for all the SNPs/datapoints and performing leave one out and leave all influential points out analyses. How is the association affected if SNPs are iteratively removed from the risk score until the heterogeneity test is no longer significant at the specified threshold? This can be performed using the `grs.filter.Qrs` function in the R package "gtx" genetics toolbox (<https://rdrr.io/cran/gtx/man/grs.filter.Qrs.html>). Likewise, how do these additional analyses affect the association between breast cancer and schizophrenia shown in Figure 2 B (which is also heterogeneous)?

Response: Thanks for the suggestions. We have added an analysis of lead SNPs by leaving all influential SNPs out and by filtering SNPs iteratively until the heterogeneity test is no longer significant. The results remained largely similar. The association between lead SNPs for schizophrenia and risk of breast cancer was somewhat attenuated, compared with the results of Figure 2A, but still significant. The association between lead SNPs for breast cancer and risk of schizophrenia was null in the original analysis and was not changed in this additional analysis. We have now added this to the Results and Online Methods.

Modified text:

Page 9, paragraph 2: *Considering the significant heterogeneity in the tests of association, we further visualized the associations by filtering all influential SNPs out (according to Cook's distance) and by removing SNPs iteratively until the heterogeneity test was no longer significant. The results remained largely similar. The association between lead SNPs for schizophrenia and risk of breast cancer was significant ($P=0.024$), although somewhat attenuated (Supplementary Figure S2).*

Page 23, paragraph 2: *In addition, to assess the effect of influential SNPs, we calculated the Cook's distance for each SNP and those with Cook's distance larger than $N/4$ were removed. We further plotted the association by removing SNPs iteratively until the heterogeneity test was no longer significant. This procedure was done by using `grs.filter.Qrs` function in the `gtx` package in R.*

Supplementary Figure S2. Associations of lead SNPs (after filtering for outlier or heterogeneity) for schizophrenia with the risk of breast cancer, and vice versa. Orange dots, variants of $P < 0.05$ after FDR adjustment. OR, odds ratio; SE, standard error; SNPs, single nucleotide polymorphisms; Phet, P for heterogeneity.

What can we learn from the SNPs contributing to the heterogeneity (see for example the tool MR-TRYX from Cho, Hemani, et al)?

Response: As far as we have understood, the MR-TRYX framework is used to identify alternative horizontal pleiotropic pathways and discover other risk factors for disease in Mendelian randomization analysis. Since we do not claim causation on the genetic association between schizophrenia and breast cancer, we did not further pursue the analysis using MR-TRYX. We, however, are certainly willing to reconsider our position on the editor's request.

- Figure 1 B indicates that the association between the breast cancer PRS and schizophrenia only really appears when large numbers of SNPs with weak breast cancer associations or even SNPs not associated with breast cancer are included in the breast cancer PRS. This seems very strange and incompatible with a "bi-directional" relationship. Can the authors perform additional analyses to prove that this is not an artefact? Even the observational estimates for association of breast cancer with subsequent schizophrenia in Table 1 are quite weak this taken together with the genetics further weakens the case for bi-directionality. How would Figure 1 B be affected by control analyses (say prostate/ovarian cancer PRS -> schizophrenia and breast cancer PRS -> psychiatric disease other than schizophrenia – all these data sets are publicly available)?

Response: Thanks for raising this issue. We agree with the reviewer that the genetic association we observed is not bidirectional (i.e., not causal). Please also refer to our response above for clarification.

We however performed additional analyses as the Reviewer suggested. Using latest GWASs of prostate cancer and bipolar disorder (*Schumacher FR, Nature Genetics, 2018, 50:928-36; Stahl EA, Nature Genetics, 2019, 51:793-803*), we observed a similar pattern for prostate cancer PRS in prediction of schizophrenia and for breast cancer PRS in prediction of bipolar disorder (i.e., the associations only appear when large numbers of SNPs are included in PRS). This pattern may partly be explained by the polygenic architecture in many psychiatric disorders, particularly in schizophrenia that is imparted by many common variants with individually small effects (*Sullivan PF, Cell, 2019, 177:162-83*). When a large number of SNPs were included in the PRS analysis, those “small” contributors significantly improve the prediction for risk of schizophrenia. Of note, the genetic correlation between breast cancer and bipolar disorder is 0.08 (SE=0.03, P = 0.02), and the genetic correlation between prostate cancer and schizophrenia is 0.05 (SE=0.03, P=0.09). Both are far lower and less significant than the genetic correlation observed between breast cancer and schizophrenia.

We have now added this observation to the Results. We did not include these additional analyses as they are beyond the scope of the present study. We are however willing to reconsider our position on the reviewer’s or editor’s request.

Modified text:

Page 8, paragraph 1: Notably, the association between the breast cancer PRS and schizophrenia appears when large numbers of SNPs weakly associated with breast cancer are included in the analysis, in support of the polygenic architecture in schizophrenia.

Figure. Polygenic risk score analysis for an alternative psychiatric disorder or cancer. PRS, polygenic risk score.

2. Given the use of genetic correlations and PRSes in this manuscript, the latent causal variable method of O'Connor and Price (PMID: 30374074) may further inform the dissection of the relationship between schizophrenia and breast cancer.

Response: Thanks for this suggestion. We agree with the reviewer this method may provide further information in dissecting the causal relationship. This paper has already calculated the genetic causality proportion (GCP) between schizophrenia and breast cancer as 0.29 (standard error (SE) 0.45) (see Supplementary Table S12 in the paper "*Distinguishing genetic correlation from causation across 52 diseases and complex traits*" PMID: 30374074) using a smaller GWAS of breast cancer. We have discussed their findings in our original Discussion: "Another study based on GAME-ON data does, however, not support a causal genetic correlation between the two diseases³³".

We however performed such analysis by using the GWAS of breast cancer we used in the present study and yielding a similar GCP (0.20, SE 0.48). Since we are not primarily aimed to determine or claim causality between these two traits, we did not include this analysis in the manuscript for now. We are however willing to reconsider our position on the editor's request.

REVIEWERS' COMMENTS:

Reviewer #1 (Remarks to the Author):

The authors have addressed the issues I raised in my comments to them. Thank you for the opportunity to review this manuscript.

Reviewer #2 (Remarks to the Author):

The authors have adequately addressed my questions and I have no further concerns or comments.